# Application of 1D ResNet for Multivariate Fault Detection on Semiconductor Manufacturing Equipment [note 1]

**DOI:** 10.3390/s23229099

**Published:** 2023-11-10

**Authors:** Philip Tchatchoua, Guillaume Graton, Mustapha Ouladsine, Jean-François Christaud

**Affiliations:** 1LIS, CNRS, Aix Marseille University, University of Toulon, 13007 Marseille, France; guillaume.graton@lis-lab.fr (G.G.); mustapha.ouladsine@lis-lab.fr (M.O.); 2STMicroelectronics, 13106 Rousset, France; 3Ecole Centrale de Marseille (Centrale Méditerranée), 13013 Marseille, France

**Keywords:** fault detection, raw sensor data, multivariate time series, semiconductor manufacturing, deep learning

## Abstract

Amid the ongoing emphasis on reducing manufacturing costs and enhancing productivity, one of the crucial objectives when manufacturing is to maintain process tools in optimal operating conditions. With advancements in sensing technologies, large amounts of data are collected during manufacturing processes, and the challenge today is to utilize these massive data efficiently. Some of these data are used for fault detection and classification (FDC) to evaluate the general condition of production machinery. The distinctive characteristics of semiconductor manufacturing, such as interdependent parameters, fluctuating behaviors over time, and frequently changing operating conditions, pose a major challenge in identifying defective wafers during the manufacturing process. To address this challenge, a multivariate fault detection method based on a 1D ResNet algorithm is introduced in this study. The aim is to identify anomalous wafers by analyzing the raw time-series data collected from multiple sensors throughout the semiconductor manufacturing process. To achieve this objective, a set of features is chosen from specified tools in the process chain to characterize the status of the wafers. Tests on the available data confirm that the gradient vanishing problem faced by very deep networks starts to occur with the plain 1D Convolutional Neural Network (CNN)-based method when the size of the network is deeper than 11 layers. To address this, a 1D Residual Network (ResNet)-based method is used. The experimental results show that the proposed method works more effectively and accurately compared to techniques using a plain 1D CNN and can thus be used for detecting abnormal wafers in the semiconductor manufacturing industry.

## 1. Introduction

Semiconductor manufacturing is a batch multi-step process, where silicon wafers undergo a sequence of complex and lengthy processing operations involving a large number of recipes and equipment types, during which electronic circuits are gradually crafted to create functional integrated circuits. Products are organized into batches of 25 silicon wafers throughout equipment production. Finalized wafers are obtained after several months of extensive processing cycles, representing hundreds of operations. The semiconductor manufacturing process is nonlinear and can be disrupted by various factors, such as equipment aging, cleaning, and repairs; the state of the wafers and wafer transfer; and preprocess chambers and chamber warm-up. As a result, there is process variability within a wafer (intrawafer variability), between wafers (interwafer variability), within a batch (intrabatch variability), and between different batches (interbatch variability). The equipment’s data, which are automatically collected by numerous sensors located on the process equipment, provide direct information about the process conditions, such as temperature, pressure, gas flow, power, capacitance, etc. This results in a vast amount of sensor data that are routinely collected and stored on appropriate media.

Modern manufacturing industries use cutting-edge big data technologies and innovative machine learning techniques to reduce manufacturing costs and improve production quality by extracting insightful knowledge from the collected data to enhance process automation, predictive analyses, and effective equipment monitoring [1]. As for equipment monitoring, its main purpose is to identify abnormalities and faults in manufacturing process operations. In manufacturing industries, equipment monitoring can be segmented into four main parts: fault detection, fault identification and diagnosis, estimation of fault magnitudes, and product quality monitoring and control [2]. The methods used to implement this monitoring are divided into three main categories: qualitative model-based, quantitative model-based, and data-driven methods [3]. In order to use model-based monitoring methods, the structure and behavior of the monitored system and all of its components must be thoroughly known and understood. Model-based monitoring methods are very reliable, but they suffer from numerous flaws, as the detailed analytical descriptions needed for their implementation are either unavailable for complex industrial processes or greatly time-consuming to obtain due to the need for extensive human intervention. Unlike model-based methods, data-driven methods do not require any a priori knowledge about the system. The models are constructed by relying solely on available process data, through which the characteristics of the system are extracted.

To guarantee consistent, continuous, and reproducible production quality, the sensor data collected from hundreds of equipment variables are utilized for equipment monitoring purposes, such as fault detection, fault diagnosis, prognosis, equipment health management, predictive maintenance, and virtual metrology. The early detection and precise classification of faulty wafers that result from abnormal processing are crucial for controlling operations, minimizing yield losses, and preventing defective wafers from progressing to the subsequent stages for each equipment. This paper, in particular, emphasizes the use of sensor data for fault detection and classification (FDC) in the semiconductor industry.

As time goes on, the strong technological push provides improved data storage and data analysis capabilities, resulting in the collected sensor data being significantly larger as the number of data samples and dimensionality jointly increase. With this increase in sensor data availability, the collected data disclose many subtleties, such as incompleteness, high dimensionality, infrequent labeling, and severely unbalanced samples. This paper focuses on high dimensionality and severely unbalanced data.

Firstly, the intricate nonlinear interactions between the signals (multiple intervariable correlations) in the high-dimensional sensor data make the detection of abnormal measurements exceedingly challenging. For data-driven tasks, it is crucial to extract solely the pertinent information, especially when dealing with multidimensional data [4]. Various feature extraction and dimensionality reduction methods have been developed to extract relevant features by performing nonlinear mappings of input data into an embedded representation [5]. The learned embedded representation contains useful features, which can be used to perform fault detection with statistical control charts or machine learning methods, resulting in improved reliability. To detect faults through feature extraction and dimensionality reduction, several unsupervised machine learning methods have been proposed based on factor analysis embedding, locally linear embedding, and singular value decomposition (SVD) embedding.

Secondly, industrial faults rarely occur, resulting in severely unbalanced data samples, where faulty samples are scarce. The rare occurrence of faults makes it difficult to constitute a dataset sufficiently balanced for effective supervised machine learning. While numerous feature extraction and classification approaches for fault detection and analysis have been presented, the fault classification accuracy remains unsatisfactory due to the severely unbalanced data samples [6]. This imposes a great limitation on the usage of supervised learning methods for fault detection. The scarcity of faulty samples has led to the widespread use of self-supervised learning methods based on Principal Component Analysis (PCA) [7], Independent Component Analysis (ICA) [8], and Partial Least Squares (PLS) [9], which can be combined with supervised learning methods such as support vector machine (SVM) [10] and k-Nearest Neighbors (k-NN) [11] for fault identification.

The high volume of data poses a challenge for machine learning methods that require extensive data preprocessing, leading to performance limitations [12]. To address this challenge in the semiconductor industry, deep learning algorithms that can handle large volumes of data without extensive preprocessing have been explored for fault detection. Additionally, deep learning algorithms can adapt and learn from new data, making them suitable for dynamic environments, where data patterns may change over time. Deep learning approaches have performed very well across a wide range of applications, effectively transforming high-dimensional information into new embedded representations with robust and meaningful characteristics. Self-supervised deep learning methods, such as stacked [13], denoising [14], convolutional [15,16,17,18], and recurrent autoencoders [19,20], have been used to enable this efficient translation of input data to embedded characteristics. Deep learning methods achieve equally good fault detection performance when working on unbalanced datasets with supervised learning methods based on Convolutional Neural Networks (CNN) [21,22,23]. In their study, Hsu et al. [22] notably used data augmentation with a sliding window to generate numerous subsequences from multiple time series, which helped avoid overfitting on the unbalanced datasets.

With the increase in dataset sizes for complex data characteristics, such as those found in multivariate time series, deeper models are needed. Deep learning models provide more accurate results as the number of layers increases. In order to achieve the most accurate models on very large datasets, the depth of the models must be continuously increased to cope with the increase in dataset sizes. However, despite being the primary method with state-of-the-art performance, deep learning techniques face the issue of vanishing/exploding gradients when the network becomes very deep. As a result, shallow counterparts may outperform deep networks [24,25]. He et al. [24] proposed residual networks (ResNet) to efficiently overcome vanishing gradients. To perform bearing fault detection, Qian et al. [26] used a ResNet classifier with model-based data augmentation to cope with the requirement of large amounts of data. This paper addresses the gradient vanishing problem in a plain 1D CNN-based fault detection method trained with a substantial amount of multivariate time series data from a semiconductor manufacturing process. To overcome this observed issue with vanishing gradients, this paper introduces a novel ResNet architecture for fault detection on multidimensional time series. The proposed architecture uses 1D convolutions, which capture both the temporal dynamics and spatial correlations in the multivariate time-series data. The approach’s effectiveness is demonstrated by analyzing two datasets and comparing them to the state-of-the-art methods. This study is an extended analysis of a work previously presented at a conference [27]. providing new and interesting insights into gradient analysis, detailed data, and fault-type description, as well as discussing detection performance for each fault type.

The remainder of this paper is organized as follows. Section 2 introduces the representative deep learning methods used in fault detection. Section 3 exposes the gradient vanishing problem and describes the proposed ResNet model. Section 4 presents the experimental setup, and Section 5 discusses the detection performance on real and simulated data from a semiconductor manufacturer. Finally, Section 6 concludes the paper and discusses future studies.

## 2. Deep Learning Methods for Fault Detection

This section introduces the nature of the sensor data and briefly presents the neural network approaches used for the experimental analysis. The gradient vanishing problem on deep CNNs is formalized, and the theory behind residual connections is explained.

### 2.1. Multivariate Time Series

A multivariate time series, also known as multidimensional time series, is a sequence of vectors that involves multiple variables recorded over a period of time, with each vector representing the state of a monitored variable at a specific time point. In other words, it is a collection of time series, where each time series corresponds to a different feature or dimension. A multivariate time series *S* with *T* time steps and *M* variables is represented as $S=[S_{1},S_{2},\ldots ,S_{T}]$, where $S_{k}=\left(s_{1,k},s_{2,k},\ldots ,s_{M,k}\right)$ is an M-dimensional vector that represents the values of the *M* variables at time *k*. In contrast to a univariate time series, which involves only a single variable, a multivariate time series can capture the relationships and interactions between multiple variables.

In the semiconductor industry, equipment sensor data are collected at a given frequency, and this can vary from one equipment to another. Sensor data variables, also referred to as status variable identification (SVID), can be collected every 1 s, 0.5 s, 0.2 s, and so on, and this value is fixed for specific equipment and never changes. Semiconductor manufacturing is a batch-processing industry, and the equipment sensor data are collected as three-dimensional data. They constitute a multivariate time series, which can be represented in a 3D matrix form, i.e., wafer number, SVID, and processing time, as shown in Figure 1.

For each SVID, all the wafers are recorded for different durations due to variations in the processing time for different recipes, as well as the time-varying behaviors inherent in semiconductor manufacturing. This leads to a non-stationary dynamic in the multivariate time series. Consequently, all the durations need to be synchronized and preprocessed to a fixed length prior to fault detection. Given the various operating conditions, there are differences in the statistical characteristics of the collected time series between one wafer and another, and one batch and another.

### 2.2. Supervised Deep Learning for Fault Detection

Long Short-Term Memory (LSTM) is a type of recurrent neural network (RNN) that is capable of processing sequential data such as time-series data by preserving information over a longer period of time compared to traditional RNNs, which suffer from the vanishing gradient problem. LSTM [28] is introduced as a solution to the vanishing gradient problem in RNNs. Instead of a single hidden state, LSTM uses a cell state and three gates (input gate, forget gate, and output gate) to control the flow of information. The cell state acts as a memory unit that can store information over longer periods of time. The gates regulate how much information is allowed to flow into or out of the cell state at each time step, allowing the LSTM to selectively forget or remember information from the past. These gating mechanisms allow LSTM to selectively remember or forget information over long periods of time, making it well suited for modeling time-series data.

LSTM is well suited for tasks such as time series classification and anomaly detection because it can learn complex temporal patterns and capture long-term dependencies and multivariate correlations in the data. For anomaly detection, the LSTM model is then trained using normal system data to learn the normal behavior of the system. Once the model is trained, it is used to detect anomalies in the system data. Anomalies are detected by comparing the output of the LSTM model for a given input with the expected output based on the model training data and computing a corresponding anomaly score. LSTM can be trained using backpropagation over time [29], which allows it to learn from past data and make predictions about future data. In [30], the authors proposed LSTM-AD, a self-supervised anomaly detection method based on stacked LSTMs. By leveraging the power of stacked LSTMs, LSTM-AD captures complex temporal dependencies in the normal time-series data. This enables it to effectively learn and predict expected behavior, making it robust against variations and anomalies in the analyzed time series. The utilization of prediction errors and thresholds allows LSTM-AD to accurately identify and flag any deviations from the learned normal patterns, providing a reliable anomaly detection mechanism. The same main author later proposed EncDec-AD in [19], an LSTM-based encoder–decoder approach for multi-sensor time-series anomaly detection. EncDec-AD reconstructs time series in reverse, uses the reconstruction error to compute anomaly scores, and sets a decision boundary threshold using the mean and standard deviation. This threshold helps classify the time-series data as either normal or anomalous. The encoder–decoder architecture of EncDec-AD is derived from a particular type of neural network: autoencoder.

Autoencoders (AEs) are a type of neural network used for unsupervised feature learning [13]. They can be used for a variety of tasks, such as data compression, image denoising, and anomaly detection. By rebuilding the input at the output, AEs approximate the identity function by reconstructing the input data as accurately as possible. They can capture complex patterns and relationships from many data types. AEs handle high-dimensional data efficiently, making them suitable for multivariate time-series analysis. An autoencoder consists of two main components: an encoder and a decoder. The encoder maps the input data to a lower-dimensional representation, whereas the decoder maps the lower-dimensional representation back to the original input. During training, the autoencoder is optimized to minimize the reconstruction error, which is the difference between the input and the output of the decoder. The denoising autoencoder (DAE) is a variant of AEs that is specifically designed to remove noise from input data. It works by training the AE to reconstruct clean versions of corrupted input data, thereby learning to extract meaningful features and patterns from noisy data, making it more resilient to noise and improving its generalization capabilities. The DAE’s robustness to input noise makes it valuable in applications where noise is prevalent.

Time-series classification using autoencoders involves training an autoencoder on a set of time-series data and then using the learned representation for classification. The encoder of the autoencoder can be thought of as a feature extractor, which maps the time-series data to a lower-dimensional feature space. The extracted features can then be used as input to a classifier, such as a support vector machine (SVM) or a random forest, to perform classification. Anomaly detection using autoencoders involves training an autoencoder on a set of normal time-series data and then using the learned representation to detect anomalies in new time-series data. Anomalies are detected by comparing the reconstruction error of the autoencoder for a given time-series data point with a threshold value. If the reconstruction error is above the threshold, the data point is considered to be an anomaly. In [15], the authors proposed using convolutional sparse autoencoders (CSAE-AD) and the corresponding convolutional denoising sparse autoencoders (CDSAE-AD) to create a self-supervised FDC approach. With the use of convolutional kernels and the addition of a sparsity penalty [31] based on the Kullback–Leibler divergence [32] in the cost function, convolutional sparse autoencoders differ considerably from basic autoencoders. CSAE-AD allows the model to learn hierarchical features from the input data and encourages the activation of only a few neurons, resulting in more efficient and robust representations. CDSAE-AD, the denoising component, further enhances performance by training the model to reconstruct clean data from noisy inputs, improving its ability to handle real-world data with noise. Later, the authors of [14] introduced an FDC approach based on stacked denoising autoencoders to extract noise-resistant features and accurately classify semiconductor data.

However, like any machine learning technique, their performance is highly dependent on the quality of the data and the specific problem being solved. Self-supervised learning methods based on LSTMs or AEs for fault detection on semiconductor time-series data perform worse than supervised learning methods, as shown in [33]. In [33], CNN-based fault detection methods exhibited the best performances.

Convolutional Neural Networks (CNNs) are a type of deep learning model commonly used in computer vision applications, but they can also be applied to time-series data. CNNs [34] are composed of multiple layers, including convolutional layers, pooling layers, and fully connected layers. Convolutional layers are the core building blocks of CNNs and consist of multiple filters that slide over the input data to extract features. The pooling layers downsample the output of the convolutional layers, reducing the dimensionality of the data. Finally, the fully connected layers are used to classify the input data. CNNs have been shown to be effective for fault detection in time-series data. The approach involves using the 1D convolutional layer to learn relevant features from the time-series data. The convolutional layer slides a kernel over the input data to extract local features, which can then be combined to form global features that are used for classification. Lee et al. [21] introduced FDC-CNN, a supervised anomaly detection approach that demonstrated good classification performance in fault detection on a Chemical Vapor Deposition (CVD) process dataset, which consisted of multivariate time series. FDC-CNN utilizes convolutional kernels to sweep the time axis of the two-dimensional input and extract both the temporal and spatial relationships between variables during feature extraction. Subsequently, Kim et al. [35] presented a modified version of FDC-CNN, which incorporates a self-attention mechanism into a CNN to improve the fault detection accuracy on an etch-process dataset. The self-attention mechanism [36] assigns attention weights via a probability distribution to different time steps, enabling the detection method to disregard irrelevant parts and concentrate on the relevant parts of a sequence, thus enhancing its ability to detect subtle anomalies.

Deep neural networks are used to enhance performance on big datasets rather than on shallow ones. Although the CNN-based methods proposed by [21,35] achieved some great results on our small datasets, they faced the vanishing gradient problem when the networks became very deep. The vanishing gradient problem is a well-known issue that can occur when training deep neural networks, including CNNs and RNNs. The vanishing gradient problem occurs when the gradients used to update the weights of a neural network during training become very small, making it difficult for the network to learn. This can happen in deep neural networks with many layers, where the gradients must pass through multiple layers during backpropagation. The gradients can become small, as they are multiplied by the weight matrices in each layer, leading to a problem where the early layers of the network learn much more slowly compared to the later layers. In time-series fault detection with CNNs, the vanishing gradient problem can occur because the input data are high-dimensional and have complex temporal dependencies. The CNN model must learn to extract relevant features from the data, and these features can be spread across multiple layers of the network. If the gradients become very small as they pass through the layers, the early layers of the network may not be able to learn the relevant features, leading to poor performance. In our case, as seen in Figure 2, there was a gradual decrease in the training and test errors as the number of layers in the CNN-based fault detection model increased from two to nine layers. From 11 layers and beyond, the training and test errors increased as the number of layers increased, resulting in a drop in the detection performance as the number of layers in the network increased. This highlights the necessity of proposing a method capable of addressing the vanishing gradient problem.

### 2.3. Residual Connections in Deep Neural Networks

He et al. [24] brought attention to the problem of performance degradation observed when CNNs deepen. As the network depth increases, the network performance begins to saturate and finally degrades. This phenomenon is caused by the vanishing gradient of deep neural networks rather than overfitting [25]. This can make it difficult for the network to learn from the training data, as the updates to the parameters based on the gradient can become insignificant. Thus, slow convergence or even complete failure to converge can be observed during the training of the network. Several network designs, including ResNet [24], Highway Network [37], and DenseNet [38], have been proposed to address this issue. All these networks share the same design principle, commonly referred to as shortcut, skip, or residual connections. Shortcut connections are a technique used in deep neural networks to accurately address the vanishing gradient problem. They allow the gradient to be directly propagated from one layer to another, bypassing intermediate layers that may cause the gradient to become small. This helps to alleviate the vanishing gradient problem and allows the network to learn more efficiently, even when it contains many layers.

In the ResNet architecture, the shortcut connections are mainly used in two ways: they can either perform identity mapping, such as in the identity block in Figure 3a, or execute a linear projection, as in the convolution block in Figure 3b. The output of the identity blocks is combined with the output of the stacked layers, which does not add any extra parameters or computational complexity to the network. Consequently, they have the same number of parameters, depth, and width, making them simple to compare to the corresponding plain networks. For an input *x*, their output *y* is defined as:(1)$$y=\sigma \left(F\left(x,\left\{W_{i}\right\}\right)+x\right)$$
where $F\left(x,\left\{W_{i}\right\}\right)$ is the residual mapping to be learned and σ is the ReLU activation function. The function $F\left(x,\left\{W_{i}\right\}\right)$ denotes several convolutional, normalization, and activation layers, where element-wise addition is executed on two feature maps, channel by channel. In convolution blocks, the shortcut connections conduct a linear projection to align the dimensions between the input *x* and the residual mapping $F\left(x,\left\{W_{i}\right\}\right)$. This linear projection is achieved by using a 1 × 1 convolutional layer with appropriate filters. By doing so, the dimensions of the input and the residual mapping are made compatible, allowing for element-wise addition. This technique helps preserve important information while enabling the network to learn more complex representations. The output of this block is:(2)$$y=\sigma \left(F\left(x,\left\{W_{i}\right\}\right)+W_{s}x\right)$$
where $W_{s}$ is a square matrix performing the linear projection of *x*. The linear projection is employed when a modification in dimension arises in the stacked layers of a block. The structure of the residual blocks is adaptable, as depicted in Figure 3, where the blocks contain two convolutional layers. However, it is possible to have additional layers and diverse configurations.

The shortcut connection allows the gradient to flow directly from the output of the residual block to the input, bypassing the convolutional layers. This helps prevent the gradient from vanishing as it propagates through the network, making it easier to train deeper models. By adding the input to the output, the network is able to learn residual functions that represent the difference between the input and the output. This makes it easier for the network to learn the underlying function being modeled, especially when the function has many complex features.

Shortcut connections have been shown to be effective in a variety of deep neural network architectures [39]. Their ability to address the vanishing gradient problem and improve training efficiency has made them an essential tool for building deep neural networks. They have helped advance the state-of-the-art in tasks such as image classification, object detection, natural language processing, and semantic segmentation.

## 3. Proposed Method for Fault Detection

In the semiconductor industry, ResNet architectures have recently been used for wafer defect detection and classification [40,41]. They aim to sort defective chips by analyzing images of wafer surfaces. In the literature, no works have addressed fault detection on multivariate time series using residual networks. This section discusses a fault detection method based on a ResNet architecture that uses 1D convolutions to process raw multivariate time series from semiconductor manufacturing equipment.

In this paper, we implement standard CNNs and CNNs with shortcut connections to convey the advantages of adding residual connections to improve the feature learning capability of deep convolutional networks on time series. Also, a ResNet-type architecture is suggested for fault detection. ResNet, an enhanced version of the standard convolutional network, is utilized to minimize training difficulty by efficiently using shortcut connections to prevent the gradient from vanishing as it propagates through the deep network. The ResNet architecture consists of a succession of residual blocks for feature extraction, followed by fully connected layers for classification.

In standard CNNs, the receptive field is a square matrix of weights that links the input layer to the convolutional layer. With a size smaller than the input data, the receptive field moves across its horizontal and vertical axes with a predetermined stride to perform convolutions. For an input *x* of size M×K, the output of a convolution operation with no padding is stored in a node and can be expressed as follows:(3)$$\begin{array}{c} y_{ij}=\sigma \left(\sum_{m=1}^{F}\sum_{n=1}^{F}w_{m,n}x_{\left(m+iS),(n+jS\right)}+b\right),\;\mathrm{for}\;0\leq i\leq \frac{M-F}{S},\;\mathrm{and}\;0\leq j\leq \frac{K-F}{S}, \end{array}$$
where *F* represents the size of the square receptive field; *S* is the stride; $x_{\left(m+iS)(n+jS\right)}$ is the input element at position (m+iS,n+jS); $w_{m,n}$ and b are the weights at position (m,n) and the bias, respectively; and σ is a nonlinear activation function, typically a rectified linear unit (ReLU). The receptive field or filter used to create a feature map contains a single-weight matrix, which means all the nodes in a feature map share the same weights. This allows the receptive field to search for a common characteristic (such as a single intervariable correlation in multivariate sensor signals) across the entire input data [21].

However, the conventional square receptive field of CNNs is not ideal for extracting intervariable and temporal correlations among all the SVIDs, which is crucial for fault detection in multivariate time-series data. To address this, the proposed architecture utilizes a rectangular receptive field that moves only along the time axis. One-dimensional (1D) convolution layers are tailored to implement this feature, operating along a single axis. For an input wafer *x* of size M×K, which represents *M* SVIDs and *K* time steps, the output of the first convolution operation with no padding, immediately after the input layer for a node, is given by:(4)$$\begin{array}{c} y_{i}=\sigma \left(\sum_{m=1}^{F}\sum_{n=1}^{M}w_{m,n}x_{\left(m+i\times S),(n\right)}+b\right),\;\mathrm{for}\;0\leq i\leq \frac{K-F}{S}, \end{array}$$
where *F* and *S* are the row size and stride length of the receptive field, respectively. The proposed approach for fault detection combines a feature extractor based on a ResNet for feature learning with a fully connected layer. The ResNet-based architecture proposed in this study includes both identity blocks (Res-block *a*) and convolution blocks (Res-block *b*) to enhance the feature extraction process, as shown in Figure 4.

The entire architecture, as illustrated in Figure 4, has some specificities. The batch normalization layer is utilized to reduce the computational complexity of the training process. The spatial dropout layer [42] is implemented to regularize the network weights and prevent overfitting. Residual blocks are employed to mitigate the degradation problem and extract distinctive features from the dataset, with two types of blocks: identity and convolution. The convolution layers in the blocks follow two design rules: (i) when the feature map size is the same, the layers have the same number of filters, and (ii) when the feature map size is halved, the number of filters per layer is doubled. The halving or downsampling is accomplished using convolution layers with a stride of 2. The pooling layer is used to reduce the dimension of the intermediary algebraic elements, which are then flattened to obtain the 1D dimension required by the fully connected layers.

The fully connected layers, also known as dense layers, form a multi-layer perceptron, which takes a one-dimensional array obtained from the output of the feature extractor. The fully connected layers are responsible for learning the complex relationships between the features extracted by the previous layers and generating the final output probabilities for each class. In this fault detection approach, the fully connected layers perform binary classification to determine if an input sample is normal or faulty.

The complexity of a control system is similar to that of a controlled system. To alleviate the complexity of the ResNet model, the number of sensors employed at each stage of a process is chosen by experts on the basis of domain knowledge. Reducing the number and quality of sensors thus helps mitigate the complexity of the monitoring algorithm. Another way of reducing the complexity of the monitoring algorithm is by using raw sensor data. Raw sensor data have no signal processing or filtering applied to them before ingestion by the ResNet, as it is not essential for its operations. Hence, signal processing can be totally skipped with no impact on the performance of the ResNet.

When using a considerably complex algorithm like ResNet for monitoring a multivariate multi-stage process, special effort has to be made during the design and training processes to ensure proper working of the algorithm owing to its complexity. The complexity of the ResNet lies in the structure of the residual blocks and the depth of the overall model. He et al. [24] proposed a set of simple rules for the design of efficient residual blocks, as described above. The suitable depth needed depends on the size of the training dataset and is determined through empirical experimentation. One model is designed, trained, and implemented according to the production recipe of a given equipment. The design changes between two models for two production recipes occur mainly in the modulation of the input layer so as to accommodate the length of the time series and the number of sensors.

## 4. Experimental Setup

This section reports a comprehensive empirical study for fault detection in multivariate time series. First, the datasets used for experimental evaluation are introduced. Then, the experimental setup and architecture details of the networks are described. Finally, the metrics are defined to analyze and discern the results obtained. The results of the proposed method are compared to the most recent findings in the literature for fault detection in the semiconductor industry.

### 4.1. Data Preparation

This paper examines the effectiveness of the proposed model using two datasets provided by STMicroelectronics Rousset 8″ fab. To simplify the analysis, we focus on one equipment and one recipe for each dataset. The raw data consist of time series with three-dimensional information (wafer, variables, and time) but are represented as a two-dimensional matrix with processing time and SVID axes only. Sensor data are collected every second for both datasets. Equipment faults are rare in semiconductor manufacturing, and the number of faulty samples available for a given production recipe is very low compared to the number of normal samples. This leads to a situation of imbalanced data, as there are more normal samples than faulty samples in training and testing datasets. The ideal composition for supervised classification is an equal number of samples for each class. Encountering unbalanced datasets is a real challenge and adds complexity to the fault detection methods.

The first dataset was obtained from a process simulator developed by STMicroelectronics that mimics the dynamics of real variables, such as the gas flow, pressure, and temperature of an etch tool. For a single recipe lasting an average of 150 s, 11 variables are monitored for a total of 7000 wafers, including 5000 normal samples and 2000 faulty samples. This results in a ratio of 28.6% faulty data, which is considered good given the rarity of faulty data in the semiconductor industry. STMicroelectronics has identified five recurrent fault types in its manufacturing processes. The first dataset comprises five distinct fault types, each with an equal distribution of 400 samples. For each fault type, faults are introduced in one process step, and they occur on at least two different variables but not concurrently. The step in which the fault occurs is randomly selected for each fault type, ensuring that faults do not occur systematically in the same step. Figure 5 portrays these five common fault types that occur during wafer manufacturing. Fault 1 represents a breakage point, creating a deviant cycle with an amplitude range ranging between 30 and 50% of the time-series maximum value for at least 10 time steps. Fault 2 represents a temporary change in value with a return to a regular level after several time steps. For fault 2, the amplitude ranges between 10 and 30% of the time-series maximum value for at least 8 time steps. Faults 3 and 4 are analogous to additive noise and sinusoidal disturbances, respectively, acting as innovational outliers that induce a trend change. Fault 3 has an amplitude ranging from 1 to 10% and occurs for at least 5 time steps. For fault 4, the amplitude ranges from 1 to 5% with damping and phase shift factors and occurs for at least 8 time steps. Fault 5 represents a peripheral point, which is an independent data point that is notably outlying, resulting from a sudden rise in value (a peak), with an amplitude between 40 and 60% of the time-series maximum value. The primary objective is to identify all types of faults, and the current study does not examine the classification of the detected faults.

The second dataset is from a plasma etching tool. The production recipe consists of a series of nine steps and lasts 130 s on average. Due to operating conditions, the time series does not have the same length from one wafer to another. The input data need to have a fixed length to be processed by neural networks. In order to have a fixed length for the dataset, the times series are all padded to a fixed length of 140 s. Among all the collected SVIDs, domain engineers selected a set of 25 for fault detection and classification. For one month of production, this represents 516 wafers, with 423 normal samples and 93 faulty samples. The ratio of faulty wafers is 18.0%, showing a class imbalance. This second dataset has only one fault type, the temporary change.

### 4.2. Neural Network Configurations

Two versions of the ResNet are proposed as fault detection methods: a ResNet with average pooling (ResNet-1) and a ResNet with spatial pyramid pooling [43] (ResNet-2). Two different pooling methods are used here for performance optimization. For comparison purposes, six neural network models are considered as benchmarks: two CNN-based, two LSTM-based, and two autoencoder-based, with different sequence encoding methods. These architectures have achieved consistent results when used for fault detection in the semiconductor industry [14,15,21,35]. The baseline methods used are stacked autoencoders (SAE-1), convolutional autoencoders (SAE-2), standard CNN (CNN-1), CNN with self-attention (CNN-2), stacked LSTM (LSTM-1), and LSTM with self-attention (LSTM-2). CNN-2 and LSTM-2 correspond, respectively, to CNN and LSTM architectures with a self-attention layer replacing the final pooling layer. SAE-1 corresponds to a stacked autoencoder, which is composed of two symmetrical artificial neural networks in a bottleneck form. SAE-2 corresponds to convolutional autoencoders composed of a symmetrical convolutional encoder and deconvolutional decoder.

To optimize the models, various configurations were evaluated for each of the previously presented models, and only the best parameters were retained to produce the final results. The neural network architectures proposed for the experimental setting were implemented using the widely acclaimed Tensorflow software, version 2.10.

The best ResNet-1 architecture comprises one convolutional layer (with 64 filters, a kernel size of 3, and a stride of 1), one spatial dropout layer (with a rate of 10%), four residual blocks (two identity blocks with 64 filters, followed by one convolution block and one identity block with 128 filters), one average pooling layer (the pool size being fixed to 2), one dense layer (with 100 units), and one dropout layer (with a rate of 10%). For ResNet-2, the architecture is the same as that of ResNet-1 with the average pooling layer replaced with a spatial pyramid pooling layer (with 32, 16, 8, 1 bins). Spatial pyramid pooling [43] maintains the spatial information in the local spatial bins. The number of bins and their size are fixed, thus generating a fixed-length representation regardless of input size.

SAE-1 is a fully connected layer-based model, comprising an encoder and a decoder network composed of dense layers, with the decoder being the mirrored version of the encoder. The encoder has three hidden layers with 22, 15, and 10 nodes. SAE-2 is a convolutional-based model, comprising an encoder, a decoder, and one dense layer (with 100 units) for classification. The decoder is a mirrored version of the encoder with deconvolutions. The encoder has three hidden layers with 44, 30, and 20 filters. The ReLU function is used as the activation function. CNN-1 is configured as follows: 11 convolutional layers with 64, 64, 64, 64, 64, 128, 128, 128, 128, 256, and 256 filters coupled with batch normalization; ReLU activation and spatial dropout (rate: 10%) layers; one dense layer (with 100 units); and one dropout layer (rate: 10%). For CNN-2, self-attention mechanism-based Luong-style attention is used. For the LSTM architecture, two layers with 128 LSTM cells each are used. In addition, one dropout layer (the rate being 10%) and one dense layer (with 100 units) are used for classification. For LSTM-2, Luong-style attention is applied. In terms of the activation function, the underlying nonlinearity in the data is enforced through the sigmoid function for the LSTM-based models.

The neural network configurations are summarized in Table 1.

### 4.3. Other Configurations

Data partitioning: The dataset is split into training and test sets with a ratio of 80–20%. This process is performed through a stratified fivefold cross-validation partitioning in order to avoid biased results. In terms of implementation, the partitioning is carried out using Scikit-learn.Weight initialization: The initial weights are defined using Glorot uniform distribution. No layer-weight constraints are set on the weight matrices for the learning process.Weight optimization: The Adam optimizer is used for the training, with the learning rate fixed at 0.0005 for all models. After numerous optimization tests, the batch sizes are, respectively, fixed at 32 for the ResNet-based, CNN-based, and autoencoder-based models and at 16 for the LSTM-based models. For all of the models, the number of epochs is fixed at 300 with early stopping, and the cost function is the binary cross-entropy.

### 4.4. Evaluation Metrics

The evaluation metrics are the *F*-scores for model efficiency assessment and the computational complexity. The *F*-score is a function of the *Precision* and the *Recall*. In this specific framework, the *Precision* (see (Equation 5)) is the ratio of actual faults among the total detected faults and the *Recall*, as detailed in (Equation 6), corresponds to the ratio of the actual faults with respect to the correct predictions.
(5)$$\mathrm{Precision}=\frac{T_{P}}{T_{P}+F_{P}},$$
(6)$$\mathrm{Recall}=\frac{T_{P}}{T_{P}+F_{N}},$$

Given (Equation 5) and (Equation 6), the *F*-score is expressed in (Equation 7). $F_{weighted}$, as expressed in (Equation 8), is used as the main score, which is a weighted sum of $F_{0}$ and $F_{1}$ that takes into account the imbalanced dataset. It follows: (7)$$F_{\beta }=\frac{(1+\beta^{2})\mathrm{Precision}.\mathrm{Recall}}{\beta^{2}\mathrm{Precision}+\mathrm{Recall}},$$
(8)$$F_{weighted}=\frac{F_{1}A_{P}+F_{0}A_{N}}{A_{P}+A_{N}},$$
where $T_{P}$, $F_{P}$, $F_{N}$, $A_{P}$, and $A_{N}$ represent true positive, false positive, false negative, actual positive, and actual negative, respectively.

The efficiency of a given model is an increasing function of the score, i.e., the model is considered very precise when the score is high (close to 1, which is the maximum upper bound).

Remark: It is worth highlighting that accuracy, which is the most intuitive way to evaluate classification models, is not a convenient efficiency measure for an imbalanced dataset [44]. This is why the $F_{weighted}$ score is proposed as the evaluation metric.

## 5. Results and Discussion

This section presents and discusses the results obtained on both a simulated and a real dataset from semiconductor manufacturing.

### 5.1. Gradient Analysis

In Figure 6, the training and validation errors during the training of shallow and deeper networks for both plain and ResNet architectures are compared. In Figure 6a, the gradual decrease in the training error from the 7-layer to the 11-layer plain network and the sudden degradation of the 13-layer plain network, which had higher training errors throughout the training process, can be observed. The same cannot be said for the validation error of the plain networks, as shown in Figure 6b. It is hypothesized that the deep plain networks may have encountered optimization difficulties and thus exponentially low convergence rates, which affected how well the training error was reduced. On the other hand, it can be seen that despite the increase in depth, the residual networks exhibited equal training errors from 7- to 13-layer networks, indicating high convergence rates. This implies that the degradation problem was adequately handled. There were gains in the detection capability from the increased depth until the 13-layer network, where degradation can be observed in both plain and ResNet architectures. This suggests that even for residual networks, there is a maximal depth beyond which performance starts to degrade. Table 2 shows the detection performance achieved for layer depths varying from 7 to 13 layers for plain and ResNet architectures. It can be seen that when comparing networks with equal depth, residual networks always demonstrated better capabilities than their plain counterparts. Subsequently, the best plain and ResNet architecture (11 layers) was retained for comparison with other deep learning-based fault detection methods. For the dataset used, training was carried out on 7 to 13 layers because networks with fewer than 7 layers are not relevant in a ResNet architecture, and all networks with more than 13 layers suffer from the vanishing gradient problem.

### 5.2. Results Analysis

Table 3 and Table 4 present the results of the proposed method and the aforementioned deep learning-based fault detection methods on the simulated and real datasets, respectively. The best results for the fault detection models, obtained with optimized hyperparameter values (best values for the learning rate, batch size, number of epochs, and dropout obtained by testing several configurations), are indicated in bold.

For both datasets in Table 3 and Table 4, the proposed ResNet-based approaches outperformed the other baseline methods by a significant margin and exhibited the best performance with the highest $F_{weighted}$ scores (0.9389 and 0.9708). The LSTM-based methods achieved the worst scores and significantly underperformed compared to the other methods. Both models achieved null values for $F_{1}$ and $F_{weighted}$ due to the inability of the models to converge on both datasets. The standard CNN-based method (CNN-1) exhibited the second-best performance on the simulated dataset, closely followed by the convolutional autoencoder (SAE-2), and both methods outperformed the fully connected-based stacked autoencoder (SAE-1) model and the attention CNN (CNN-2), which achieved the worst results among the converging methods. Nonetheless, on the real dataset, the convolutional autoencoder exhibited the second-best performance with only a slightly better $F_{weighted}$ score (<1%) compared to the standard CNN (CNN-1). On the real dataset, the standard CNN (CNN-1) and the convolutional autoencoder (SAE-2) performed better than the attention CNN (CNN-2) and the fully connected-based stacked autoencoder (SAE-1) by a rather large margin (>10%). In addition, the margin was very tight on the simulated dataset (<1%). For all models, the $F_{0}$-score was always better than the $F_{1}$-score by a significant margin, which provides insights into the detection capacities of the models.

The $F_{0}$-score measures the ability of the models to correctly identify normal samples, whereas the $F_{1}$-score evaluates their ability to identify faults. The results demonstrate that all models were generally effective in identifying normal samples, with $F_{0}$-scores consistently above 0.8. The proposed model achieved the highest scores of 0.9600 and 0.9825 on the simulated and real datasets, respectively. Even though it is important to correctly identify normal samples, fault identification is the critical factor, and the $F_{1}$-score is a more informative performance metric. The LSTM-based models failed to encode lengthy sequences over time, resulting in null scores for $F_{1}$ on both datasets.

With our task being fault identification, we focus on the $F_{1}$-score. For all models, the $F_{1}$-score was lower than the $F_{0}$-score, with the proposed model achieving the best $F_{1}$-scores of 0.8865 and 0.9189 on the simulated and real datasets, respectively. This suggests that the models struggled more to identify faults than normal samples and the degree of difficulty varied among the models, as indicated by the difference between the two scores. This difference was significant (>10%) for all models, except for the proposed ResNet models, highlighting their superiority and establishing them as a reliable FDC method. All models struggled to identify faults because of the unbalanced dataset used for training the models, with fewer faulty samples than normal ones.

Table 5 presents the results of the best ResNet method (ResNet-1) and the best CNN-based fault detection method (CNN-1) for each fault type on the simulated dataset. The results here focus on the $F_{1}$-score only to determine how well the methods identified the different fault types as faults. In Table 3, it can be seen that the overall $F_{1}$-scores were 0.8315 for CNN-1 and 0.8865 for ResNet-1, which do not provide information on how these methods performed in detecting each fault type. The results in Table 5 show that the best proposed ResNet-1 performed better than the best CNN-based method for each fault type, with some remarkable performance gaps. For faults 3 and 4, it can be seen that CNN-1 performed poorly compared to the other fault types. Faults 3 and 4, which are illustrated in Figure 5, refer to noise and sinusoidal disturbances, respectively. For fault 3, when looking at the noise distribution (Gaussian normal distribution centered on 0), a large number of samples were close to zero most of the time. Moreover, the amplitudes of the noise faults were quite small (see Section 4.1), which made them more difficult to detect because they appeared as recurrent industrial disturbances rather than faults. With the data being raw time series, differentiating fault types 3 and 4 from simple industrial noise disturbances was more difficult for the two models. This was even more true for fault type 3, where even the proposed ResNet-1 struggled, although it exhibited better performances compared to the plain CNN-1. Regarding the nature of the noise disturbances, despite being less pronounced compared to those of the other four fault types, the detection results obtained using the proposed ResNet-1 were quite good. One of the reasons for implementing deeper networks, as in the proposed ResNet-1, is to craft a method capable of effectively detecting all fault types.

## 6. Conclusions

This paper proposes a ResNet-based fault detection method for semiconductor process monitoring using multivariate sensor signals. The proposed model redesigns the first convolutional layer to consider the structural characteristics of raw multidimensional sensor data and extract meaningful correlation and temporal information. The use of residual blocks with shortcut connections improves training and mitigates the degradation problem of deep neural networks, resulting in better fault detection performance. The proposed model is evaluated using both simulated and real data from the semiconductor industry, outperforming state-of-the-art and baseline models for fault detection. All five fault types addressed in this study are successfully detected, with the proposed method achieving the best detection performance for each. This study also demonstrates that residual networks outperform their plain counterparts with equal layer depths. The small size of the real dataset used for training and testing does not significantly impact the generalizability of the conclusions.

Future work will focus on adapting the model to work with variable-length sensor data and providing insights for fault diagnosis. Research will also be conducted to enable the model to detect faults in multiple recipes with a single model and classify detected faults based on their nature, proposing relevant elements for equipment root-cause diagnosis.

## Figures and Tables

**Figure 1 sensors-23-09099-f001:**
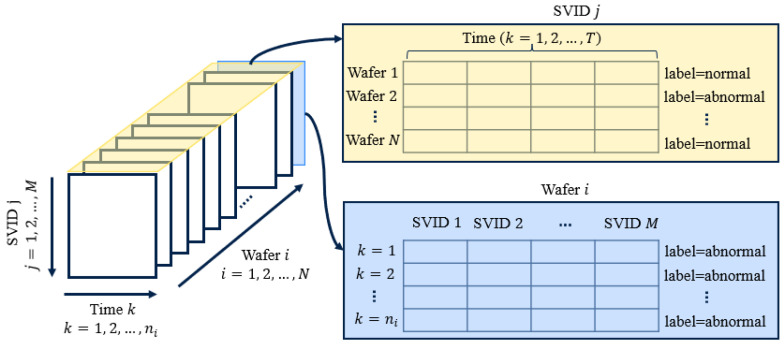
Multidimensional sensor data representation. A labeled 3D data matrix for *N* wafers, *M* SVIDs, and with varying process times $n_{i}$ per wafer.

**Figure 2 sensors-23-09099-f002:**
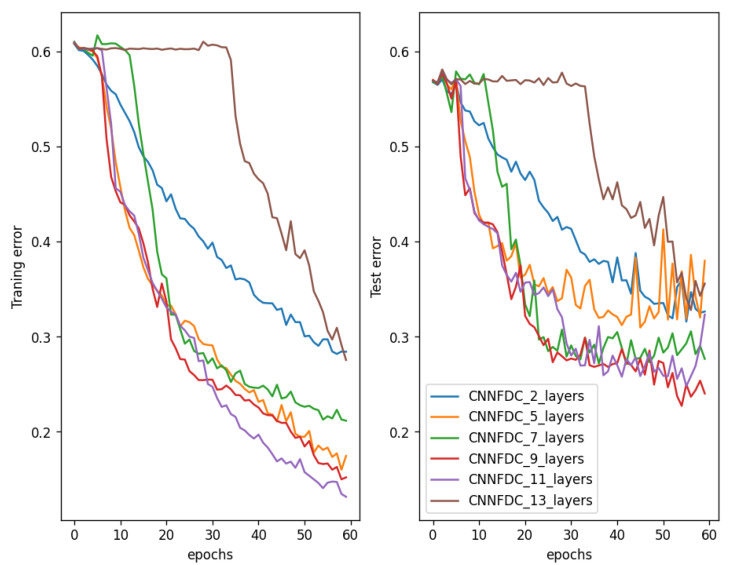
Training errors (**left**) and test errors (**right**) of plain CNNs with 2, 5, 7, 9, 11, and 13 layers on a multivariate time-series dataset. The training and test errors gradually decreased as the networks deepened but started to increase from the 11-layer network, confirming the vanishing gradient problem on deeper networks.

**Figure 3 sensors-23-09099-f003:**
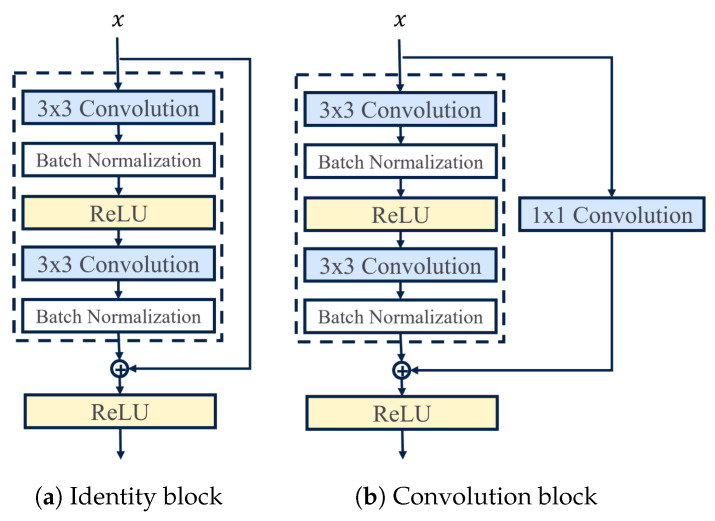
The two residual block structures (with shortcut connection) behind the ResNet architecture proposed in [24].

**Figure 4 sensors-23-09099-f004:**
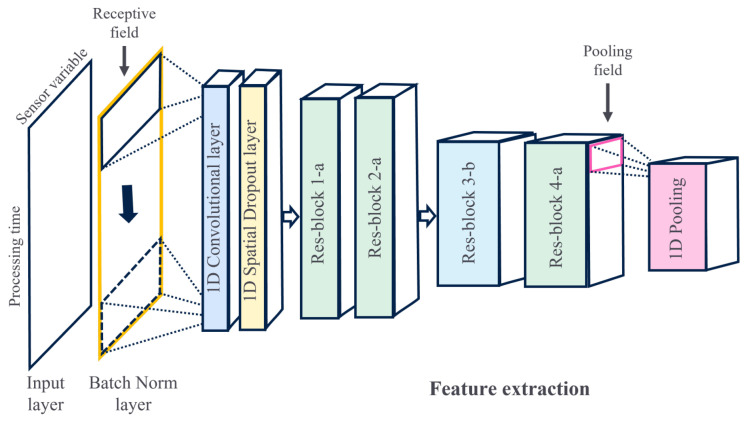
ResNet-based feature extraction. Res-blocks *x*-*a* are the identity blocks shown in Figure 3a, and Res-blocks *x*-*b* are the convolution blocks shown in Figure 3b.

**Figure 5 sensors-23-09099-f005:**
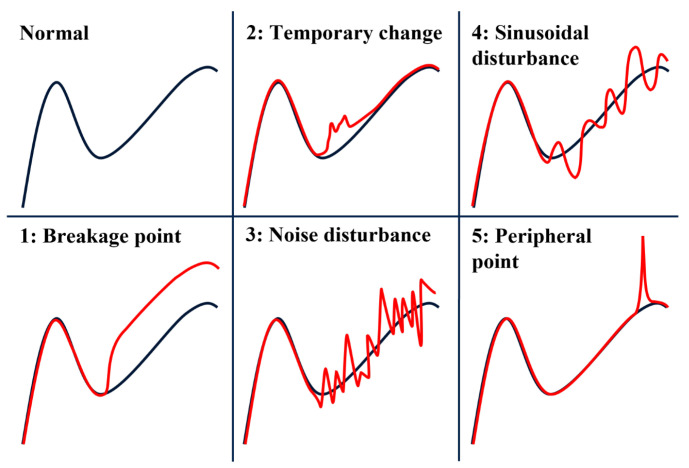
Description of the 5 common fault types (in red). These anomalies transpire across different variables and can be either atomic or aggregate in nature. Atomic anomalies involve abnormal values for a single variable, whereas aggregate anomalies arise from groups of variables deviating collectively.

**Figure 6 sensors-23-09099-f006:**
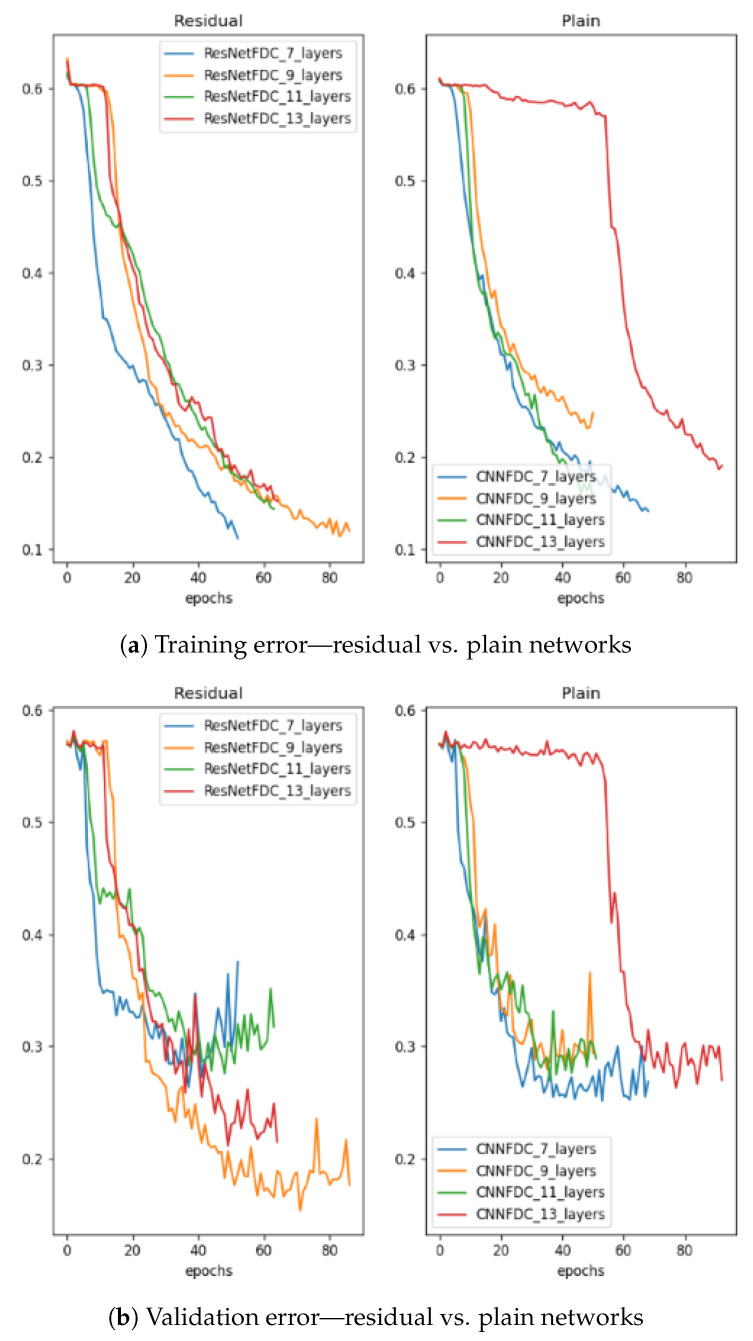
Training and validation on the simulated dataset presented in Section 4. In these plots, the residual networks have no extra parameters compared to their plain counterparts.

**Table 1 sensors-23-09099-t001:** Summary of the neural network configurations of the various methods used for fault detection on the simulated dataset.

Hyperparameter	ResNet-1	ResNet-2	CNN-1	CNN-2	LSTM-1	LSTM-2	SAE-1	SAE-2
Model specificity	Residual blocks	Residual blocks	Plain blocks	Self-attention CNN	Stacked LSTM	Self-attention LSTM	Stacked autoencoders	Conv autoencoders
Number of feature extraction layers	11	11	11	2	2	2	6	6
Activation function	ReLU	ReLU	ReLU	ReLU	Sigmoïd	Sigmoïd	ReLU	ReLU
Number of classification layers	2	2	2	2	2	2	1	2
Pooling before classification	Average pooling	Spatial pyramid pooling	Average pooling	No pooling	No pooling	No pooling	No pooling	No pooling
Batch size	32	32	32	32	16	16	32	32

**Table 2 sensors-23-09099-t002:** Fault detection performance of residual vs. plain networks with the same layer depth on the simulated dataset.

Method	Layers	$F_{0}$	$F_{1}$	$F_{\mathrm{weighted}}$
ResNet-7	7	0.9507	0.8607	0.9250
Plain-7	7	0.9371	0.8058	0.8996
ResNet-9	9	0.9589	0.8837	0.9374
Plain-9	9	0.9414	0.8187	0.9063
ResNet-11	11	**0.9599**	**0.8865**	**0.9389**
Plain-11	11	0.9425	0.8315	0.9108
ResNet-13	13	0.9518	0.8567	0.9246
Plain-13	13	0.9367	0.8097	0.9004

**Table 3 sensors-23-09099-t003:** Fault detection performance on the simulated dataset.

Method	$F_{0}$	$F_{1}$	$F_{\mathrm{weighted}}$
**ResNet-1**	0.9599	**0.8865**	**0.9389**
ResNet-2	**0.9600**	0.8855	0.9387
CNN-1	0.9425	0.8315	0.9108
CNN-2	0.9229	0.7370	0.8698
LSTM-1	0.8333	-	-
LSTM-2	0.8333	-	-
SAE-1	0.9276	0.7715	0.8830
SAE-2	0.9412	0.8260	0.9083

**Table 4 sensors-23-09099-t004:** Fault detection performance on the real dataset.

Method	$F_{0}$	$F_{1}$	$F_{\mathrm{weighted}}$
**ResNet-1**	**0.9825**	**0.9189**	**0.9708**
ResNet-2	0.9651	0.8333	0.9410
CNN-1	0.9659	0.8125	0.9379
CNN-2	0.9239	0.4167	0.8312
LSTM-1	0.8995	-	-
LSTM-2	0.8995	-	-
SAE-1	0.9153	0.5161	0.8423
SAE-2	0.9714	0.8485	0.9490

**Table 5 sensors-23-09099-t005:** F1-scores for fault detection performance for each fault on the simulated dataset.

Method	Fault 1	Fault 2	Fault 3	Fault 4	Fault 5
**ResNet-1**	**0.9873**	**0.9610**	**0.6885**	**0.9160**	**0.9873**
CNN-1	0.9200	0.8072	0.5664	0.5789	0.9682

## Data Availability

Restrictions apply to the availability of these data. Data was obtained from STMicroelectronics and are not available.

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
