# Peer review of "Application of 1D ResNet for Multivariate Fault Detection on Semiconductor Manufacturing Equipment†"

_sensors, 2023, doi:10.3390/s23229099_

Round 1

Reviewer 1 Report

Comments and Suggestions for Authors

The gradient vanishing problem is established on a simple 1D CNN-based fault detection algorithm in this research, and a novel ResNet architecture for fault detection on multidimensional time series that captures both temporal dynamics and spatial correlations is proposed. The paper represents good work as well as it is well written. However, in my opinion, the paper in question needs revision before it can be considered for publication in the Sensors journal. Some things to think about are as follows:

1-     The introduction section should be extended to include more recent papers especially in the years 2022 and 2023.

2-     The authors did not clearly mention what the novelty of this work is compared with the existing literature.

3-     Make sure that all the symbols in the equations are defined since there is no nomenclature section. 

4-     The findings of this study should be compared to the most recent findings in the literature. The authors should compare the results obtained from the presented study with those of existing studies in the literature.

5-     The conclusion section should be enhanced and extended.

6-     The conclusions should contain a qualitative comparison between the presented method and other methods already studied in the literature.

7-     More references in the years 2022 and 2023 should be cited to be up to date. Only four references in the years 2020-2023 are cited (zero references in the 2023 year, one reference in the 2022 year, one reference in the 2021 year and two references in the 2020 year).

Comments on the Quality of English Language

The English language should be checked throughout the whole paper.

Author Response

  • More references of the recent years (2021, 2022, 2023) have been added so as to include the latest works on ResNet and their usage for fault detection, with a focus on the semiconductor industry. This has been done in sections 1 and 3.
  • The novelty of the study is also added in section 1

Reviewer 2 Report

Comments and Suggestions for Authors

My comments are as follows :

1. Abstract need to be rewritten. Specifically, line 13-14 do not reflect the correct meaning of sentence. Further, the keywords related to methodology should be included.

2. Similar title with same authors is already published with minor changes. How the published conference paper is different from submitted manuscript. Refer to the link :

https://ieeexplore.ieee.org/document/9853997

3. Author's specific contribution wrt literature review should be specifically mentioned. It should also highlight the novelty in terms of exploration of new algorithms, optimization techniques, detailed investigation etc.

4. When signals are captured,to filter the noise signal processing techniques need to be applied which is a standard practice.Here no filtering applied on captured signal.

5. It is always a good practice to rely on ten-fold cross validation prediction results instead of training and testing results. Kindly discuss/apply in revised version along with following references :

a. https://doi.org/10.3390/batteries9020125

b. https://doi.org/10.1080/13658816.2017.1346255

6. Authors mentioned that dataset is imbalanced. Further, authors have not applied any filtering techniques. In such situation, how to rely on classification results.

7. The ML/DL algorithms applied by authors seems to be frequently used. Kindly explore the possibility of utilization of new algorithms for fault classification.

8. Conclusion section need to include key numerical values obtained from various classifiers so that reader can understand the results in a better way.

Comments on the Quality of English Language

English language is ok.

Author Response

  • The novelty of the proposed method and the contribution of the paper with respect to the literature has been added and highlighted in the introduction.
  • The justification for no signal processing is added in section 3.
  • The usage of 5-fold cross validation is mentioned in section 4.
  • The origin of the imbalance in the datasets is added in section 4. The imbalance in data resulting from a specificity of semiconductor manufacturing where faulty samples are rare.

Reviewer 3 Report

Comments and Suggestions for Authors

The topic of the paper and the result is interesting. However, some points must be clarified, and some result presentations must be modified in order to be accepted as a publication. Please kindly consider the following suggestions.

  1. From the explanation in the introduction, it is not clear what is the weaknesses of the existing related research that become the motivation for this research. Please explain it.
  2. Please explain more clearly the contribution of this paper in the introduction section.
  3. Please improve the quality of Figure 6 (clarity, resolution, font size)
  4. For the result in Fig 6, please explain why the authors only simulate 7-13 layer network in this paper.
  5. Please modify the presentation of results in Tables 2 - 4. By only looking at the table, it is unclear which one is the proposed method and which is the previous study’s result. Please also cite the references for the previous study’s results.
  6. Please summarize the neural network configuration in the table so the reader can understand the big picture of the configuration easily without reading the whole paragraphs.
  7. When explaining the dataset, please also explain the imbalanced data. Currently, there is no explanation for the imbalanced data condition.

Author Response

  • The weaknesses of the existing related research are further explained in section 1, notably their inability to have good performances on large datasets of complex data structures as multivariate time series.
  • The novelty of the proposed method and the contribution of the paper is also highlighted in the introduction.
  • The reason for using 7 to 13 layers for the gradient analysis is given in section 5 and the size of the figure 6 increased.
  • The origin of the imbalance in the datasets is added in section 4. The imbalance in data resulting from a specificity of semiconductor manufacturing where faulty samples are rare.
  • A table summarizing the neural network configurations of the various methods used in the evaluation is also added in section 4 for the big picture.

Reviewer 4 Report

Comments and Suggestions for Authors

The reviewed work studies a multivariate algorithm for identification of malfunction in industrial semiconductor equipment. My observations thereon are provided below:

1. The first question is a conceptual one: when using a considerably more complicated algorithm for controlling a multi-stage process, would it not be necessary to also make a special effort to ensure proper working of the algorithm itself (owing to its complexity)? Theoretically, the complexity of the control system may approach that of the controlled system. Where is the optimum in relation to the control system complexity? This subject needs proper commenting.

2. The second question is also conceptual: how does the number and quality of the sensors employed at each stage of the process affect the monitoring algorithm operation? How does one understand whether or not their data are satisfactory? This is also necessary to discuss.

3. The proposed algorithm is presented as universal and applicable to any industrial semiconductor process. Nevertheless, such processes may be widely different. Are there any peculiarities of the algorithm specific to different processes? A comment to this effect would be highly desirable.

4. The presented algorithm mainly relies on the data from sensors registering predominantly external parameters. However, problems may also arise from internal defects of the used materials, both semiconductor and those of the fabrication line. How could the proposed algorithm monitor this? A comment is in order.

If the Authors address the provided questions in a revised version of the manuscript, it may be published in Sensors.

Author Response

  • In section 3, two paragraphs on the complexity of the control system with respect to the controlled system is added. This aims at clarifying and explaining how these complexities are dealt with in our proposed approach.
  • The impact on the choice of the number and quality of sensors use is explained in section 3.
  • The proposed algorithm is generic. There are no peculiarities specific to different processes other than conforming the input layer to the input size of a given process.
  • The proposed algorithm aims at detecting equipment failure or dysfunctioning. Defects linked to materials are detected by other means in semiconductor manufacturing.

Round 2

Reviewer 1 Report

Comments and Suggestions for Authors

Some issues have been addressed by the authors. However, certain concerns were not addressed.

Comments on the Quality of English Language

Minor editing of English language required

Author Response

Please find point-to-point reply in attached file.

Reviewer 2 Report

Comments and Suggestions for Authors

My observations based on revised version and authors reply :

1. First, there is no significant difference between the title of already published literature and title of submitted manuscript.

2. Authors have not addressed comments individually. I have not seen any comments reply.

3. There is no response/justifications related to signal processing techniques,10-fold cross valdiation etc.

Comments on the Quality of English Language

English language is ok.

Author Response

(The authors gave the same response as above.)

Reviewer 4 Report

Comments and Suggestions for Authors

In response to my observations, important information was added to the manuscript that made it more interesting and comprehensible. My comments have been fully addressed by the Authors in the revised manuscript, which may be now published.

Author Response

.

Round 3

Reviewer 1 Report

Comments and Suggestions for Authors

The authors have addressed all the points.

Comments on the Quality of English Language

 Minor editing of English language required

Author Response

Dear Reviewer,

I have read the manuscript and can't spot out where the English needs to be improved. Can you please highlight the sections or paragraphs to be edited ?

Thank you.

Best regards.

Reviewer 2 Report

Comments and Suggestions for Authors

After reading second revised version, my comments are as follows:

1. Still, the title of conference paper and submitted manuscript are almost same. If there is novelty/extension/modifications why it is not reflected in manuscript.

2. Authors not able to justify the comments 2 and comments 3. Still there is no novelty and significant extension observed in  revised version.

3. Not agree for response to comment 4. Signal processing or any implementations should not be a choice, rather it is a standard methodology.In industry ML need to be deploy and there are high chances that classification results are biased.

4. Further authors have not implemented and apply 10-fold CV which removes any kind of biasedness in methdology.Further when 10-fold CV is implemented prediction results are reliable due to utilization of whole dataset for training and testing.

Comments on the Quality of English Language

English language is ok.

Author Response

Dear Reviewer,

Here are my responses to each your comments : 

  1. Is there any obligation for the titles of the conference paper and that of the submitted manuscript to be completely different ? I haven't heard of such a rule. Further more, the two papers addressing a common issue, with the submitted manuscript being an extension of the conference paper, is it not normal that there is some similarity between the two papers ? Also, the novelty/extension/modifications of the submitted manuscript with respect to the conference paper has already been addressed in the previous responses.
  2. As said in the previous responses, the novelty of the submitted manuscript lays in the new interesting insights on :

     - Gradient analysis, showing how the gradient vanishing problem manifests itself on the used data and how it is addressed ;

     - Data description on the nature of faults present, their amplitude and their occurrence ;

     - Additional state-of-the-art deep learning algorithms for comparison purposes of the proposed approach ;

     - Detection performances per fault type, giving further information on how the proposed method deals with the different fault types in the data.

  3. As previously stated, for the purpose of our research, no signal processing is done on the data on the demand of our industrial partners. Furthermore, signal processing being the standard methodology doesn't makes it mandatory for Machine/Deep Learning. The results of the current study, as many other methods in the literature, prove that signal processing isn't compulsory for a good, unbiased modelisation. 
  4. In the previous comments, we stated the fact that due to the limited data size and the unbalanced classes, 5-fold cross validation has been preferred to 10-fold cross validation. In the design and implementation of the study leading to the submitted manuscript, 5-fold cross validation has been used.

I have read the manuscript and can't spot out where the English needs to be improved. Can you please highlight the sections or paragraphs to be edited ?

Thank you.

Best regards.